# A functional framework for nonsmooth autodiff with *maxpooling* functions

**Bruno Després**                                    *bruno.despres@sorbonne-universite.fr*
*Sorbonne Université, Université Paris Citée, CNRS, LJLL, F-75005 Paris, France*
*MEGAVOLT, INRIA, Paris, France*

**Reviewed on OpenReview:** *https://openreview.net/forum?id=qahoztvThX&noteId=fjpRXQIAli*

## Abstract

We make a comment on the recent contribution Boustany (2024), by showing that the Murat & Trombetti (2003) Theorem provides a simple and efficient mathematical framework for nonsmooth automatic differentiation of *maxpooling* functions. In particular it gives a the chain rule formula which correctly defines the composition of Lipschitz-continuous functions which additionnaly are piecewise-$C^1$. The formalism is applied to four basic examples, with some tests in PyTorch. A self contained proof of an important Stampacchia formula is in the appendix.

## 1 Introduction

In this work we make a comment on the recent contribution by Boustany (2024). We will show that the Murat & Trombetti (2003) Theorem gives a natural framework for nonsmooth automatic differentiation (nonsmooth autodiff). More generally we believe that the methods developed hereafter offer a comprehensive mathematical functional framework for the description of nonsmooth autodiff. Recent contributions on mathematical issues for nonsmooth autodiff are developed in Bolte & Pauwels (2021); Bertoin et al. (2021; 2023); Boustany (2024) and references therein, based on the Clarke's generalized derivative (Clarke, 1990). The historical remark in Bolte & Pauwels (2021) provides insights on related topics.

Notations and developments in this self contained article are kept to the minimum.

The first part introduces the notation and provides two examples which illustrate the apparent paradox discussed in this work. The second part focuses on a theoretical justification of chain rule manipulations for non smooth functions. Similar problems have been identified in the past in the partial differential community. Seminal references are Stampacchia (1963), Ambrosio & Dal Maso (1990) and Kinderlehrer & Stampacchia (2000). In the context of this work, the most important reference is the Murat & Trombetti (2003) Theorem which can be very conveniently adapted to describe the mathematics of nonsmooth autodiff. A preliminary work Berner et al. (2019) has already highlighted the potential interest of the Murat-Trombetti Theorem for nonsmooth autodiff. However the discussion was restricted to activation functions only, so to the best of our understanding, the scope in Berner et al. (2019) was limited to fully-connected feed-forward neural network function where the main difficulty can be ruled out by means of a trivial simplification using $R'(0) = 0$ where $R$ is the ReLU activation function. Quite surprisingly the main illustrative example in the Murat-Trombettti contribution has exactly the structure of a modern basic convolutional neural network function (CNN) (Bengio et al., 2017) with a *maxpooling* function.

That is why we provide a detailed and self contained proof of the Murat-Trombetti Theorem. The original part of this work is Theorem 2 and Proposition 1 which explicit the chain rule for composition of many Lipschitz-continuous and piecewise-$C^1$ functions. We hope that this discussion will contribute to popularize this approach among the Machine Learning scientific community and to show its potential for the description of many problems that intervene in nonsmooth automatic differentiation.

The third part is devoted to some structural properties of the set of Lipschitz-continuous functions which are piecewise-$C^1$. This set is evidently an additive algebra and, more important, it is an associative algebra for composition and nonsmooth automatic differentiation. In the fourth part, we compare with other frameworks for nonsmooth differentiation, such as Clarke's generalized differential and alike and discuss the case of differentiation with respect the the weights of the Neural Network in view of minimization of a Loss function. In the fifth part, we will describe the solution of basic problems in the context of the Murat-Trombetti Theorem. A general conclusion will be that the gradient constructed through nonsmooth autodiff in PyTorch is systematically equal to an associated gradient in the sense of Murat-Trombetti.

The author warmly thanks François Murat for his deep explanations on mathematical methods for the differentiation of nonsmooth functions. He also thanks Hervé Le Dret for valuable help in manipulating Borel sets.

## 2    Notation and examples

A fully-connected **feed-forward neural network** function $f : \mathbb{R}^{a_0} \to \mathbb{R}^{a_{\ell+1}}$ can be written as

$$f = f_\ell \circ S_\ell \circ f_{\ell-1} \circ S_{\ell-1} \circ \cdots \circ f_1 \circ S_1 \circ f_0 \tag{1}$$

where the parameter $\ell$ is identified with the number of hidden layers. The functions $f_i$ for $i = 0, \ldots, \ell$ are affine functions with varying input and output dimensions $(a_0, a_1, \ldots, a_{\ell+1}) \in \mathbb{N}^{\ell+2}$ with $a_i > 0$, for $i = 0 \ldots, \ell$. More precisely, $f_i(x_i) = W_i x_i + b_i \in \mathbb{R}^{a_{i+1}}$ for all $x_i \in \mathbb{R}^{a_i}$. The intermediate functions $S_i$ for $i = 1, \ldots, \ell$ are nonlinear activation functions (Sharma et al., 2017). We adopt the normalization that $0 \leq S_i'(x) \leq 1$ for almost all $x \in \mathbb{R}$. The ReLU function $S_i = R$

$$R(x) = \max(0, x)$$

is an extremely popular activation function used in production codes (Bengio et al., 2017). The mathematical issues discussed in this work come from the fact that the ReLU function is not differentiable at $x = 0$. The point-wise non differentiability is shared with other basic functions in Neural Networks. For example *maxpooling* used in **convolutional neural networks** (Bengio et al., 2017) is based on the maximum function

$$(a, b) \mapsto \max(a, b)$$

which is also not uniquely differentiable for $a = b$. In practice *maxpooling* is activated on blocks or windows of numbers on tensors of arbitrary dimensions. With adapted natural notations from Pintore & Després (2024), the representation (1) holds for CNN as well by taking $f_\ell$ to be the identity and $S_\ell$ to be a *softmax* function (Bengio et al., 2017).

A function $f : \mathbb{R}^a \to \mathbb{R}^b$ is Lipschitz-continuous if there exists $L \geq 0$ such that $\|f(x) - f(x')\| \leq L\|x - x'\|$ for all $x, x' \in \mathbb{R}^a$. Therefore all these the activation functions $S_i$, and by extension *maxpooling* functions and all similar functions, are Lipschitz-continuous by assumption.

Since linear functions $f_i$ are clearly Lipschitz-continuous and the activation functions described above are also Lipschitz-continuous, then the feed-forward function (1) is Lipschitz-continuous by composition, that is $f \in \mathrm{Lip}(\mathbb{R}^{a_0} : \mathbb{R}^{a_{\ell+1}})$. Any intermediate step is also Lipschitz-continuous, that is

$$f_r \circ S_r \circ \cdots \circ f_1 \circ S_1 \circ f_0 \in \mathrm{Lip}(\mathbb{R}^{a_0} : \mathbb{R}^{a_{r+1}}) \quad \text{for } 0 \leq r \leq \ell$$

as well as $S_r \circ \cdots \circ f_1 \circ S_1 \circ f_0 \in \mathrm{Lip}(\mathbb{R}^{a_0} : \mathbb{R}^{a_r})$ for $0 \leq r \leq \ell$. The Rademacher (1919) Theorem, see also Morrey Jr (2009), states that the functions $f$, $f_r$ and $S_r$ for $0 \leq r \leq \ell$ are differentiable almost everywhere (that is up to sets of zero measure). So the gradient of $f$, written as a matrix, is bounded $\nabla f \in L^\infty(\mathbb{R}^{a_0} : \mathcal{M}_{a_{\ell+1}, a_0}(\mathbb{R}))$. Similarly one has

$$A_r = \nabla f_r \circ S_r \circ \cdots \circ S_1 \circ f_0 \in L^\infty(\mathbb{R}^{a_0} : \mathcal{M}_{a_{r+1}, a_0}(\mathbb{R}))$$

and

$$B_r = \nabla S_r \circ \cdots \circ S_1 \circ f_0 \in L^\infty(\mathbb{R}^{a_0} : \mathcal{M}_{a_r, a_0}(\mathbb{R})).$$

If all functions are smooth enough, the matrix-valued functions $A_r$ and $B_r$ are defined without ambiguity. However these matrix-valued functions are ambiguous because the regularity is only Lipschitz-continuous in our case. Then a natural mathematical question is to interpret the chain the rule

$$\nabla f(x) = A_\ell(x) B_\ell(x) A_{\ell-1}(x) B_{\ell-1}(x) \dots A_1(x) B_1(x) A_0(x) \tag{2}$$

which is a product of matrix-valued functions which are ambiguously defined. For feed-forward functions, our notations imply that $A_r(x) = \nabla_{x_r}(W_r x_r + b_r) = W_r$ is a constant matrix so it is trivially non ambiguous. The difficulty is then concentrated in the matrices $B_r(x)$.

**Remark 1** (On notations). *It must be noted that nonsmooth automatic differentiation with respect to the weights and biases leads to the same difficulties. For example the function $x \mapsto f(x)$ in (1) would be noted $x \mapsto f_\theta(x)$ where the variable $\theta$ denotes all the parameters of Neural Network (typically all parameters $W_i$ and $b_i$ of the linear functions). The differentiability must be taken with respect to $\theta$.*

*Fundamental is the minimization of functions used for the identification of the parameters. Consider that one knows a finite dataset $\mathcal{D} \subset \mathcal{R}^{a_0} \times \mathcal{R}^{a_{\ell+1}}$ made with pairs of input-output variables. Then supervised learning can be formulated as the minimization problem*

$$\theta = \mathrm{argmin}_\theta J(\theta) \ where \ J(\theta) = \sum_{(x,y) \in \mathcal{D}} |f_\theta(x) - y|^2. \tag{3}$$

*With our hypotheses the function $\theta \mapsto J(\theta)$ is Lipschitz continuous, however it is known to be highly non convex. The tools developed in this work can be used to describe the differentiability properties of $\nabla_\theta J$, see Section 5.2.*

*For the sake of coherence of the notations of this work, we restrict the notations to differentiability with respect to the input variable $x$, and let the reader do the easy generalization to differentiability with respect to the parameter $\theta$.*

To illustrate the issue and the apparent paradox, we consider two examples.

### 2.1 First example

The first example is degenerate in a sense. We take $f_0(x) = w_0 x$ where the weight is $w_0 \in \mathbb{R}$ and $S_1(x) = R(x)$. Then $f(x) = R(w_0 x)$ and (2) becomes

$$f'(x) = R'(w_0 x) w_0, \tag{4}$$

where $R'(y) = 0$ for $y < 0$, $R'(y) = 1$ for $y > 0$, but $R'(0)$ is not defined. A problem shows up if $w_0 = 0$ because the function $x \mapsto R'(w_0 x)$ is not defined in this case. Of course, one can argue that the multiplication by $w_0 = 0$ is enough to obtain the correct solution $f' = 0$. However this operation is not correct on solid mathematical grounds because the function $x \mapsto R'(w_0 x)$ is not mathematically defined for $w_0 = 0$.

### 2.2 Second example

The second example is more problematic. It comes from Murat & Trombetti (2003) but is rewritten here with Neural Networks notation. Consider two functions $f_0$ and $S_1$. The function $f_0 : \mathbb{R} \to \mathbb{R}^2$ is linear

$$f_0(x) = (x, x) = W_0 x \ with \ W_0 = (1, 1)$$

while the second function $S_1$ is a *maxpooling* function over two values

$$S_1(y) = \max(y_1, y_2), \ where \ y = (y_1, y_2) \in \mathbb{R}^2. \tag{5}$$

Then $f = S_1 \circ f_0$ is the identity $f(x) = x$ for all $x \in \mathbb{R}$, so that $f' \equiv 1$ is of course bounded. The gradient of $f_0$ is $\nabla f_0 = W_0$. The gradient of $S_1$ is defined almost everywhere. There are three cases

$$\begin{cases} \text{if } y_1 > y_2 & \nabla S_1(y) = \begin{pmatrix} 1 \\ 0 \end{pmatrix}, \\ \text{if } y_1 < y_2 & \nabla S_1(y) = \begin{pmatrix} 0 \\ 1 \end{pmatrix}, \\ \text{if } y_1 = y_2 & \nabla S_1(y) \text{ is not defined.} \end{cases} \tag{6}$$

Now since $f(x) = x$, the chain rule formula (2) writes

$$1 = \nabla S_1(f_0(x))W_0. \tag{7}$$

But $f_0(x) = (x, x)$ so only the third case matters in (6) therefore $\nabla S_1(f_0(x))$ is nowhere defined. Since $W_0 \neq 0$, one can not argue as in the first example that the product with $W_0$ is enough to recover the correct solution. One obtains a paradox since the left hand side equal to 1 is perfectly known while the right hand side is not even a correctly defined function.

## 2.3 General case

In general, the chain rule formula (2) does not make sense because the matrix-valued functions in the right hand side are not defined in a non ambiguous manner. In our opinion, it is related to some seemingly erratic behavior of nonsmooth autodiff related to the Boustany (2024) example detailed in Section 6.3.

## 3 The Murat-Trombetti Theorem

The Murat & Trombetti (2003) Theorem offers a natural solution to the apparent paradox explained in the previous examples. It is written in the sequel for the context of Neural Networks. We slightly adapt the functional setting and notations from Murat & Trombetti (2003) and use two different notations for the gradient $\nabla f$ and for an associated gradient $\widetilde{\nabla} f$. The examples will show that associated gradients correspond to gradients calculated with autodiff.

We will need the notion of Lipschitz and piecewise-$C^1$ functions $\mathbb{R}^a \to \mathbb{R}^b$ which is defined as follows. Consider a finite decomposition of $\mathbb{R}^a$ in Borel sets or Borel pieces $P^\alpha$

$$\mathbb{R}^a = \bigcup_\alpha P^\alpha, \qquad P^\alpha \cap P^\beta = \emptyset \text{ for } \alpha \neq \beta$$

for finite number of values of $\alpha$, that is $1 \leq \alpha \leq \alpha_{\max} < \infty$.

For simple functions, the pieces $P^\alpha$ are usually piecewise affine with Lipschitz regularity. It means that they correspond to polygons, polyhedrons, lines, half lines, points and all finite unions and intersections of such objects in any dimension. This intuitive condition is evident in our examples so we do not detail it.

**Definition 1.** *We say that a Lipschitz-continuous function $f : \mathbb{R}^a \to \mathbb{R}^b$ is piecewise-$C^1$ if there exists Borel pieces $P^\alpha$ and functions $f^\alpha \in Lip(\mathbb{R}^a : \mathbb{R}^b) \cap C^1(\mathbb{R}^a : \mathbb{R}^b)$ for $1 \leq \alpha \leq \alpha_{\max}$ with the representation*

$$f(x) = \sum_\alpha \mathbf{1}_{P^\alpha}(x)f^\alpha(x) \quad \forall x \in \mathbb{R}^a. \tag{8}$$

*Here the notation $\mathbf{1}_\omega$ denotes the indicatrix function of a set $\omega$, that is $\mathbf{1}_\omega(x) = 1$ for $x \in \omega$ and $\mathbf{1}_\omega(x) = 0$ for $x \notin \omega$. By definition $f^\alpha(x) = f(x)$ for all $x \in P^\alpha$ and $\nabla f^\alpha \in L^\infty(\mathbb{R}^a : \mathcal{M}_{b,a}(\mathbb{R})) \cap C^0(\mathbb{R}^a : \mathcal{M}_{b,a}(\mathbb{R}))$.*

The main idea in Murat & Trombetti (2003) is the following.

**Definition 2.** *We say that the gradient associated to the representation (8) is*

$$\widetilde{\nabla} f(x) = \sum_\alpha \mathbf{1}_{P^\alpha}(x)\nabla f^\alpha(x) \in \mathcal{M}_{b,a}(\mathbb{R}) \quad \forall x \in \mathbb{R}^a. \tag{9}$$

*In brief we refer to it as an associated gradient.*

*The associated gradient is a real $b \times a$ matrix defined everywhere, that is for all $x \in \mathbb{R}^a$. It is not unique since it depends on the representation (8) which is not unique. Corollary 1 will show that it is equal to the gradient almost everywhere.*

**Theorem 1** (Murat-Trombetti). *Consider two functions. The first one $u \in Lip(\mathbb{R}^a : \mathbb{R}^b)$ is Lipschitz-continuous. The second one $v \in Lip(\mathbb{R}^b : \mathbb{R}^c)$ is Lipschitz-continuous and piecewise-$C^1$ with a representation (8) and with an associated gradient (9). Then the chain rule identity holds in $L^\infty(\mathbb{R}^a : \mathcal{M}_{c,a}(\mathbb{R}))$*

$$\nabla(v \circ u) = \widetilde{\nabla} v \circ u \, \nabla u$$

where $\widetilde{\nabla} v \circ u$ denotes the function such that $\widetilde{\nabla} v \circ u(x) = \widetilde{\nabla} v(u(x))$ for all $x \in \mathbb{R}^a$.

*Hint of the proof.* The key part of the proof is the third step in (Murat & Trombetti, 2003, page 590) that we reproduce mutatis mutandi within the convenient functional setting. The two first steps in Murat & Trombetti (2003) are evident for $u$ and $v$ Lipschitz. The fourth and fifth steps concern additional properties.

- Since $v$ is piecewise-$C^1$, it admits a piecewise smooth approximations denote as $v^\alpha \in C^1(\mathbb{R}^b)$ for all $\alpha$. Since $\nabla v^\alpha \circ u \in C^0(\mathbb{R}^a : \mathbb{R}^c)$ is a continuous function, the chain rule is applied without difficulty

$$\nabla(v^\alpha \circ u) = \nabla v^\alpha \circ u \, \nabla u. \tag{10}$$

This identity holds almost everywhere (a.e.) with respect to $x \in \mathbb{R}^b$. Let $U^\alpha$ be the set

$$U^\alpha = \{x \in \mathbb{R}^a \; u(x) \in P^\alpha\}. \tag{11}$$

Since $U^\alpha$ is the pre-image of the Borel set $P^\alpha$ by the continuous function $u$, that is $U^\alpha = u^{-1}(P^\alpha)$, then $U^\alpha$ is also a Borel set so it is a measurable set.

- One notes that

$$v(u(x)) = v^\alpha(u(x)) \quad a.e. \;\; x \in U^\alpha. \tag{12}$$

To use this identity one notes $w = v \circ u - v^\alpha \circ u$. Then one uses an intuitive but non trivial important property from Stampacchia (1963); Kinderlehrer & Stampacchia (2000) (a self contained proof is in the appendix). It writes

$$\nabla w(x) = 0 \quad a.e. \;\; x \in \{x \in \mathbb{R}^a : w(x) = 0\}. \tag{13}$$

Since $U^\alpha \subset \{x \in \mathbb{R}^a : w(x) = 0\}$, it yields using (10)

$$\nabla(v \circ u)(x) = \nabla v^\alpha \circ u(x) \, \nabla u(x) \quad a.e. \;\; x \in U^\alpha. \tag{14}$$

One also has

$$\mathbf{1}_{U^\alpha}(x) = \mathbf{1}_{P^\alpha}(u(x)) \quad a.e. \;\; x \in \mathbb{R}^a. \tag{15}$$

- Consider the difference of the two terms in the claim $A(x) = \nabla(v \circ u)(x) - \widetilde{\nabla} v \circ u \, \nabla u$. It writes also

$$
\begin{array}{rcl}
A(x) & = & \nabla(v \circ u) - \left(\sum_\alpha \mathbf{1}_{P^\alpha}(u(x))\nabla v^\alpha(u(x))\right) \nabla u(x) \\
& = & \left(\sum_\alpha \mathbf{1}_{P^\alpha}(u(x))\right) \nabla(v \circ u)(x) - \left(\sum_\alpha \mathbf{1}_{P^\alpha}(u(x))\nabla v^\alpha(u(x))\nabla u(x)\right) \\
& = & \sum_\alpha \mathbf{1}_{P^\alpha}(u(x)) \left(\nabla(v \circ u)(x) - \nabla v^\alpha(u(x))\nabla u(x)\right) \\
& = & \sum_\alpha \mathbf{1}_{U^\alpha}(x) \left(\nabla(v \circ u)(x) - \nabla v^\alpha(u(x))\nabla u(x)\right) \qquad \text{(use (15))} \\
& = & \sum_\alpha \mathbf{1}_{U^\alpha}(x) \, (0) \qquad \text{(use (14))}
\end{array}
$$

where all manipulations holds a.e. with respect to $x$. Therefore $A(x) = 0$ a.e. which is the claim. $\qquad \square$

**Corollary 1.** *The associated gradient of Definition 2 is equal to the gradient $\nabla f$ almost everywhere with respect to $x$. That is $\widetilde{\nabla} f = \nabla f$ in the space $L^\infty(\mathbb{R}^a : \mathcal{M}_{a,a}(\mathbb{R}))$.*

*Proof.* Write Theorem 1 with $v = f$ and $u(x) = x$, so that $\nabla u = I$ is the identity matrix. $\qquad \square$

We now consider the composition of Lipschitz-continuous and piecewise-$C^1$ functions, as many as desired. With respect to the literature on nonsmooth differentiation discussed in Section 5, it seems that the next result is the first of its kind since it has no equivalent neither in Clarke (1990) nor even in Murat & Trombetti (2003).

**Theorem 2.** *Consider a Neural Network function $f$ defined by the composition formula (1) where all functions $f_r$ and $S_r$ are Lipschitz for $1 \le r \le \ell$. Assume moreover that all $f_r$ and $S_r$ are piecewise-$C^1$ so that they admit associated gradients (9)*

$$\widetilde{A}_r = \widetilde{\nabla} f_r \circ S_r \circ \cdots \circ S_1 \circ f_0 \in L^\infty(\mathbb{R}^{a_0} : \mathcal{M}_{a_{r+1}, a_0}(\mathbb{R}))$$

*and*

$$\widetilde{B}_r = \widetilde{\nabla} S_r \circ \cdots \circ S_1 \circ f_0 \in L^\infty(\mathbb{R}^{a_0} : \mathcal{M}_{a_r, a_0}(\mathbb{R})).$$

*Then the chain rule can be written as a product of associated gradients*

$$\nabla f = \widetilde{A}_\ell(x)\widetilde{B}_\ell(x)\widetilde{A}_{\ell-1}(x)\widetilde{B}_{\ell-1}(x) \ldots \widetilde{A}_1(x)\widetilde{B}_1(x)\widetilde{A}_0(x) \quad a.e. \;\; x \in \mathbb{R}^a \tag{16}$$

*where the right hand side is defined for all $x \in \mathbb{R}^a$.*

*Proof.* The proof is based on iterations of Theorem 1.
• The first step is based on $f = f_\ell \circ (S_\ell \circ f_{\ell-1} \circ \ldots f_0)$. It yields

$$\nabla f = \widetilde{\nabla} f_\ell \circ S_\ell \circ f_{\ell-1} \circ \ldots \ldots f_0 \; \nabla S_\ell \circ f_{\ell-1} \circ \cdots \circ f_0.$$

The gradient $\nabla f$ is expressed as the product of one associated gradient and one gradient.
• The second step is based on $S_\ell \circ f_{\ell-1} \circ \ldots f_0 = S_\ell \circ (f_{\ell-1} \circ \ldots f_0)$. One obtains

$$\nabla S_\ell \circ f_{\ell-1} \circ \cdots \circ f_0 = \widetilde{\nabla} S_\ell \circ f_{\ell-1} \circ \ldots \ldots S_1 \circ f_0 \; \nabla f_{\ell-1} \circ \ldots S_1 \circ f_0$$

One substitutes this expression in the expression for $\nabla f$ which is now expressed as the product of two associated gradients and one gradient.
• Then iterations yields the result. The right hand side of the claim is defined for all $x$ by definition of the associated gradients. $\qquad\square$

If the functions $f_r$ are linear one can simplify using $\widetilde{A}_r(x) = W_r$ which is a constant matrix. One obtains the representation $\nabla f = W_\ell \widetilde{B}_\ell(x) W_{\ell-1} \widetilde{B}_{\ell-1}(x) \ldots W_1 \widetilde{B}_1(x) W_0$ a.e. $x \in \mathbb{R}^a$ where the right hand side is defined for all $x \in \mathbb{R}^a$.

## 4 The algebra of Lipschitz-continuous and piecewise-$C^1$ functions

We show that the set of functions which are Lipschitz-continuous and piecewise-$C^1$ functions is an algebra for the operation of composition, provided of course that the dimensions of the functions match. This property is the consequence of the next Lemma.

**Lemma 1.** *Consider two functions $u \in Lip(\mathbb{R}^a : \mathbb{R}^b)$ and $v \in Lip(\mathbb{R}^b : \mathbb{R}^c)$. Assume $u$ is piecewise-$C^1$ with a representation $u(x) = \sum_\alpha \mathbf{1}_{P^\alpha}(x) u^\alpha(x)$ for all $x \in \mathbb{R}^a$. Assume $v$ is piecewise-$C^1$ with a representation $v(x) = \sum_\beta \mathbf{1}_{Q^\alpha}(x) v^\alpha(x)$ for all $x \in \mathbb{R}^b$. Then $f = u \circ v \in Lip(\mathbb{R}^a : \mathbb{R}^c)$ is piecewise-$C^1$ with a representation*

$$f(x) = \sum_{\alpha,\beta} \mathbf{1}_{R^{\alpha,\beta}}(x) f^{\alpha,\beta}(x) \text{ for all } x \in \mathbb{R}^a, \tag{17}$$

*where $R^{\alpha,\beta} = \{x \in P^\alpha, \; u(x) \in Q^\beta\}$ and $f^{\alpha,\beta} = v^\beta \circ u^\alpha$ for all $\alpha$ and $\beta$.*

*Moreover the gradient $\widetilde{\nabla} f$ associated to the representation (17) satisfies*

$$\widetilde{\nabla} f(x) = \widetilde{\nabla} v \circ u(x) \widetilde{\nabla} u(x) \text{ for all } x \in \mathbb{R}^a. \tag{18}$$

*Proof.* One has $R^{\alpha,\beta} = P^\alpha \cap u^{-1}(Q^\beta)$. Since $u$ is continuous, then $R^{\alpha,\beta}$ is the intersection of two Borel sets, so it is also a a Borel set.

A proof that $R^{\alpha,\beta} \cap R^{\alpha',\beta'} = \emptyset$ for $(\alpha,\beta) \neq (\alpha',\beta')$ is by contrapositive. Assume $R^{\alpha,\beta} \cap R^{\alpha',\beta'} \neq \emptyset$. Then there exists $x \in \mathbb{R}^a$ such that $x \in R^{\alpha,\beta}$ and $x \in R^{\alpha',\beta'}$. Therefore $x \in P^\alpha \cap P^{\alpha'}$ so $\alpha = \alpha'$. Denote $y = u(x)$. One has $y \in Q^\beta \cap Q^{\beta'}$ so $\beta = \beta'$. So $R^{\alpha,\beta} \cap R^{\alpha',\beta'} \neq \emptyset \Rightarrow (\alpha,\beta) = (\alpha',\beta')$ which is the contrapositive.

Therefore the sets $R^{\alpha,\beta}$ are a piecewise decomposition of $\mathbb{R}^a$.

Consider the function $g(x) = f(x) - \sum_{\alpha,\beta} \mathbf{1}_{R^{\alpha,\beta}}(x) v^\beta \circ u^\alpha(x)$ and let us show it vanishes. Take $x \in R^{\alpha,\beta}$, then $g(x) = f(x) - v^\beta \circ u^\alpha(x)$. Since $x \in P^\alpha$, then $u^\alpha(x) = u(x)$. Moreover $y = u^\alpha(x) = u(x) \in Q^\beta$, so

$v^\beta(y) = v(y)$. It yields that $g(x) = f(x) - v \circ u(x)$. Since it holds for all $\alpha$ and $\beta$, then $g$ is the null function which is the first part of the claim.

The final part is shown as follows. Consider the difference $B(x) = \widetilde\nabla f(x) - \widetilde\nabla v(x)\widetilde\nabla u(x)$ where the associated gradients are defined from their corresponding representation. One has

$$B(x) = \sum_{\alpha,\beta} \mathbf{1}_{R^{\alpha,\beta}}(x)\nabla f^{\alpha,\beta}(x) - \sum_{\beta} \mathbf{1}_{Q^\beta}(u(x))\nabla v^\beta(u((x)) \sum_{\alpha} \mathbf{1}_{P^\alpha}(x)\nabla u^\alpha(x).$$

For $x \in R^{\alpha,\beta}$ then

$$\begin{aligned}
B(x) &= \nabla f^{\alpha,\beta}(x) - \nabla v^\beta(u((x))\nabla u^\alpha(x) \\
&= \nabla(v^\beta \circ u^\alpha)(x) - \nabla v^\beta(u((x))\nabla u^\alpha(x) \\
&= \nabla v^\beta \circ u^\alpha(x)\nabla u^\alpha(x) - \nabla v^\beta \circ u((x)\nabla u^\alpha(x).
\end{aligned}$$

But $R^{\alpha,\beta} \subset P^\alpha$ so $u^\alpha(x) = u(x)$. Therefore $B(x) = 0$ for $x \in R^{\alpha,\beta}$. Since it is true for all $\alpha$ and $\beta$, one obtains the second part (18) of the claim. $\qquad\square$

The next Lemma shows that the previous algebra is associative. We need some notations to state the result.

We consider three functions $u \in \text{Lip}(\mathbb{R}^a : \mathbb{R}^b)$, $v \in \text{Lip}(\mathbb{R}^b : \mathbb{R}^c)$ and $v \in \text{Lip}(\mathbb{R}^c : \mathbb{R}^d)$, all of them being piecewise-$C^1$

$$\begin{cases}
u(x) = \sum_\alpha \mathbf{1}_{P^\alpha}(x)u^\alpha(x) & x \in \mathbb{R}^a, \\
v(x) = \sum_\beta \mathbf{1}_{Q^\beta}(x)v^\beta(x) & x \in \mathbb{R}^b, \\
w(x) = \sum_\gamma \mathbf{1}_{R^\gamma}(x)w^\gamma(x) & x \in \mathbb{R}^c,
\end{cases}$$

with their corresponding associated gradient. For these functions and for the other ones below, we only write the representation of the functions since the representation of their gradient is immediate.

The function $w \circ v \circ u$ admits two formally different representations, the first one assembled in the order $w \circ (v \circ u)$, the second one assembled in the order $(w \circ v) \circ u$. It might yield two different associated gradients. Actually this not the case and both representations and their associated gradients are the same as shown in Lemma 2 for which we need some preliminary notations.

Let us firstly detail the representation (17) of $f = v \circ u$. We write it

$$f(x) = \sum_{\alpha,\beta} \mathbf{1}_{S^{\alpha,\beta}}(x)f^{\alpha,\beta}(x) \text{ for all } x \in \mathbb{R}^a$$

where $S^{\alpha,\beta} = \{x \in P^\alpha, u(x) \in Q^\beta\}$ and $f^{\alpha,\beta} = v^\beta \circ u^\alpha$. Invoking one more time Lemma 1, the function $g = w \circ (v \circ u) = w \circ f$ is Lipschitz-continuous and piecewise-$C^1$ with a representation deduced from the one of $w$ and the one of $f$. We write it

$$g(x) = \sum_{\alpha,\beta,\gamma} \mathbf{1}_{S^{(\alpha,\beta),\gamma}}(x)w^\gamma \circ f^{\alpha,\beta}(x) \tag{19}$$

where $S^{(\alpha,\beta),\gamma} = \{x \in S^{\alpha,\beta}, f^{\alpha,\beta}(x) \in R^\gamma\}$.

The second possibility is to start from $p = w \circ v$ which admits the representation (17) with an associated gradient (18). We write it

$$p(x) = \sum_{\beta,\gamma} \mathbf{1}_{T^{\beta,\gamma}}(x)p^{\beta,\gamma}(x) \text{ for all } x \in \mathbb{R}^b$$

where $T^{\beta,\gamma} = \{x \in Q^\beta, v(x) \in R^\gamma\}$ and $p^{\beta,\gamma} = w^\gamma \circ v^\beta$. Finally $q = (w \circ v) \circ u = p \circ u$ has the representation

$$q(x) = \sum_{\alpha,\beta,\gamma} \mathbf{1}_{T^{\alpha,(\beta,\gamma)}}(x)p^{\beta,\gamma} \circ u^\alpha(x) \tag{20}$$

where $T^{\alpha,(\beta,\gamma)} = \{x \in P^\alpha, u(x) \in T^{\alpha,\beta}\}$.

**Lemma 2.** *The representation of g (19) and the representation of q (20) are identical. More precisely :*

*the Borel sets are equal $S^{(\alpha,\beta),\gamma} = T^{\alpha,(\beta,\gamma)}$ for all $\alpha, \beta, \gamma$;*

*the composed functions are equal $w^\gamma \circ f^{\alpha,\beta} = p^{\beta,\gamma} \circ u^\alpha$ for all $\alpha, \beta, \gamma$;*

*and the functions are equal $g(x) = q(x)$ for all $x \in \mathbb{R}^a$.*

*The associated gradients of g and q are identical as well.*

*Proof.* Expansion of all terms shows that $S^{(\alpha,\beta),\gamma} = T^{\alpha,(\beta,\gamma)} = \{x \in P^\alpha, u(x) \in Q^\beta, v \circ u(x) \in R^\gamma\}$ and that $w^\gamma \circ f^{\alpha,\beta} = p^{\beta,\gamma} \circ u^\alpha = w^\gamma \circ v^\beta \circ u^\alpha$, so the representations are the same. The associated gradients are equal as a consequence of (18). □

The previous results are used to propose another version of Theorem 2.

**Proposition 1.** *Make the assumptions of Theorem 2. Then two properties hold:*

*function f itself is continuous-Lpischitz and piecewise-$C^1$;*

*the left hand side of (16) can be written*

$$\widetilde{\nabla} f = \widetilde{A}_\ell(x)\widetilde{B}_\ell(x)\widetilde{A}_{\ell-1}(x)\widetilde{B}_{\ell-1}(x)\ldots\widetilde{A}_1(x)\widetilde{B}_1(x)\widetilde{A}_0(x) \quad \text{for all } x \in \mathbb{R}^a \tag{21}$$

*where a representation formula for the associated gradient $\widetilde{\nabla} f$ is determined from the multiplication in any order of the associated gradients in the right hand side.*

*Proof.* Use Lemma 1 and Lemma 2. □

Note that the addition of two Lipschitz-continuous and piecewise-$C^1$ functions is also a Lipschitz-continuous and piecewise-$C^1$ function. This evident property will be used in the numerical Section.

**Lemma 3.** *Consider two functions $u, v \in Lip(\mathbb{R}^a : \mathbb{R}^b)$ where u is piecewise-$C^1$ with a representation $u(x) = \sum_\alpha \mathbf{1}_{P^\alpha}(x)u^\alpha(x)$ for all $x \in \mathbb{R}^a$ and v is piecewise-$C^1$ with a representation $v(x) = \sum_\beta \mathbf{1}_{Q^\alpha}(x)v^\alpha(x)$ for all $x \in \mathbb{R}^b$. Then $f = u + v \in Lip(\mathbb{R}^a : \mathbb{R}^b)$ is piecewise-$C^1$ with a representation*

$$f(x) = \sum_{\alpha,\beta} \mathbf{1}_{R^{\alpha,\beta}}(x)f^{\alpha,\beta}(x) \text{ for all } x \in \mathbb{R}^a,$$

*where $R^{\alpha,\beta} = P^\alpha \cap Q^\beta$ and $f^{\alpha,\beta} = u^\alpha + v^\beta$ for all $\alpha$ and $\beta$, and the associated gradients are additive $\widetilde{\nabla} f = \widetilde{\nabla} u + \widetilde{\nabla} v$.*

*Proof.* $R^{\alpha,\beta}$ is the intersection of two Borel sets, so it is also a a Borel set. The rest of the proof is evident. □

The multiplication of a Lipschitz-continuous and piecewise-$C^1$ function $f$ by a real number $\lambda \in \mathbb{R}$ gives a function $\lambda f$ which evidently is also Lipschitz-continuous and piecewise-$C^1$.

## 5 Additional theoretical considerations

This Section contains some reflexions about the positioning of the previous results with respect to the literature on non smooth differentiation. The discussion below does not pretend to be exhaustive.

### 5.1 Comparison with other nonsmooth differentiation frameworks

We base the discussion on two corpus of references, which are on the one hand the works issued from Clarke's generalized gradients (Clarke, 1975; 1990) and on the other hand the recent contribution Li et al. (2020) and references therein which compare different notion of gradients for the characterization of stationary

points in nonsmooth optimization. We restrict the discussion to functions which are Lipschitz-continuous and piecewise-$C^1$.

Non smooth optimization has a rich mathematical history concerning the development of generalized gradients beyond the classical derivative. The first notion is sub-differential for convex functions (Clarke, 1990), where typically the sub-gradient at $x = 0$ of the absolute value function $x \mapsto f(x) = |x|$ is the closed interval $\partial f(0) = [-1, 1] \subset \mathcal{R}$. This interval contains all slopes $\alpha \in \mathbb{R}$ such that $\alpha x \leq f(x)$ for all $x$. The definition of sub-differential is restricted to convex functions.

A more general notion is the generalized gradient in the sense of Clarke (Clarke, 1990), also referred to as Clarke's subdifferential. It can be shown that the Clarke's generalized gradient is the convexification of the local directions of differentiability (called the Bouligand differential), that is (Li et al., 2020)

$$\partial_C f(x) = \mathrm{conv}\left(\partial_B f(x)\right).$$

This definition applies to concave functions such as $x \mapsto g(x) = -|x|$. It has the important interest that $\partial_C g(0) = [-1, 1]$ so $0 \in \partial_C g(0)$ which is a characterization of the fact that $x = 0$ is an stationary point (i.e. extremal point) of $g$.

However Clarke's generalized gradients are not the panacea to treat all situations. A paradoxical situation taken from Li et al. (2020) is for the function $x \mapsto h(x) = x + x^2 \sin(1/x)$. The derivative is 1 for $x < 0$, and is equal to $1 + 2x \sin(1/x) - \cos(1/x)$ for $x > 0$. The classical derivative at $x = 0$ is clearly $h'(0) = \lim_{y \to x, \ y \neq x} \frac{h(y) - h(x)}{y} = 1$ while it can be checked that the Clarke's gradient is $\partial_C f(0) = [0, 2]$ which is of course not satisfactory. Also Clarke's gradient does not satisfy the sum rule but only an embedding rule

$$\partial_C(f_1 + f_2) \subset \partial_C f_1 + \partial_C f_2. \tag{22}$$

The example in Li et al. (2020) is as follows. Take $f_1(x) = \max(x, 0)$ (the ReLU function) and $f_2(x) = \min(x, 0)$. Then $(f_1 + f_2)(x) = x$ so one can check that $\partial_C(f_1 + f_2)(0) = \{1\}$ while $\partial_C f_1(0) + \partial_C f_2(0) = [01] + [0, 1] = [0, 2]$. So the embedding (22) is loose even in elementary situations. Clarke's generalized gradients of product of functions is also a loose embedding (Clarke, 1990, Proposition 2.3.13).

Another point is that Clarke's generalized gradients can be used for the composition of two functions but with restrictions. Typically at least one over the two functions must be either smooth (that is differentiable in the classical sense) or regular which boils down to be essentially either concave or convex (Clarke, 1990, Theorem 2.3.10 Chain II). Up to the knowledge of the author of this contribution, Clarke's generalized gradient does not manage to describe the gradient of the composition of more than two nonsmooth general functions. The partial-differential-equation based theory Ambrosio & Dal Maso (1990) is also restricted to the composition of two functions.

On the contrary the associated gradient in the sense of Murat-Trombetti is a function defined everywhere which is equal the gradient in the sense of distribution. This equality holds up to a set of measure zero since it holds in the sense of distribution. With respect to the examples just discussed above in the Section, striking properties are that the Murat-Trombetti itself, formula (16) in Theorem 2 and formula (21) in Proposition 1 are all equalities. The set of Lipschitz-continuous and piecewise-$C^1$ functions introduced in this work is an algebra. Of course this theoretical gain has its own price, typically an associated gradient is not unique. However the applicative examples in next Section will show that this non uniqueness can also be seen as an excellent property, because it corresponds to what is observed in non smooth automatic differentiation in standard softwares (such as Pytortch, Tensorflow, Jax, . . . ).

## 5.2   Differentiation with respect to the weights and application to minimization

All previous considerations about differentiation with respect to the input variable $x$ have immediate generalization for differentiation with respect to the parameters $\theta$ of the Neural Network, see Remark 1. The main difference concerns the sets of measure zero evoked for example either in the proof of Theorem 1 or in in Formula (16). Indeed these sets of measure zero are now to be considered as embedded in the parameter space. Since it is theoretically difficult to determine the influence of these sets in general, let us consider instead a practical scenario.

This practical scenario is a basic gradient descent method for the Loss function (3)

$$\theta'(t) = -\widetilde{\nabla} J(\theta(t)) \tag{23}$$

where we assume that the gradient of the Loss is a certain type of modified gradient (note that softwares necessarily provide a numerical gradient for all $\theta$, if not the code would not run in all situations). We write the modified gradient as

$$\widetilde{\nabla} J(\theta) = 2 \sum_{(x,y)\in\mathcal{D}} \left\langle f_\theta(x) - y, \widetilde{\nabla}_\theta f_\theta(x) \right\rangle.$$

Clearly the history of the gradient descent (23) depends on the initial guess $\theta(0) = \theta_0$ and on the choice of the modified gradient because a modified gradient is not unique.

Being able to reach any conclusion on the influence of modified gradients on training sessions is a task of formidable importance for the theoretical study of practical training sessions.

## 6    Examples

We illustrate the interest of the Murat-Trombetti Theorem on simple examples where the number of layers is limited for the sake of simplicity and the differentiation is taken with respect to a given variable which can represent either the input $x$ or the parameter $\theta$.

### 6.1   Back to the first example

The issue is the value of the derivative of the ReLU function at the origin.

One can simply use three Borel pieces $P^1 = (-\infty, 0)$, $P^2 = (0, \infty)$ and $P^3 = \{0\}$ to construct the associated derivative $\widetilde{R}'$. The three smooth functions are $f^1$, $f^2$ and $f^3$. By definition $f^1(x) = R(x) = 0$ for $x \in P^1$, then $(f^1)'(x) = 0$ for $x \in P^1$. Similarly $f^2(x) = R(x) = x$ for $x \in P^2$, then $(f^2)'(x) = 1$ for $x \in P^2$. The only constraint on $f^3$ is $f^3(0) = 0$, so $(f^3)'(0)$ can be any real number. Let us note $(f^3)'(0) = z \in \mathbb{R}$.

So the associated derivative is

$$\widetilde{R}'(x) = 0 \text{ for } x < 0, \quad \widetilde{R}'(x) = 1 \text{ for } x > 0, \quad \widetilde{R}'(x) = z \text{ for } x = 0.$$

Clearly the associated derivative depends on the choice of $z$.

Most of the studies about the influence of the derivative of the ReLU at the origin are restricted to $z = 0$ that is to $\widetilde{R}'(0) = 0$, see Boustany (2024); Berner et al. (2019); Bertoin et al. (2021). The previous detailed analysis shows that $z \neq 0$ is also possible.

Whatever the value of $z$, then (4) becomes $\widetilde{f}' = \widetilde{R}' w_0$ which is non ambiguous for $w_0 = 0$ since $\widetilde{R}'$ is correctly defined. Note that the derivative of $f$ is written as an associated gradient (associated derivative in this case) thanks to Lemma 1. However, once again, the use of an associated gradient is not mandatory in this case since there is no real difficulty for $w_0 = 0$.

### 6.2   Back to the second example

To construct an associated gradient for the *maxpooling* function $S_1$ (5), one can distinguish three Borel pieces which are $P^1 = \{(y_1, y_2) \in \mathbb{R}^2 \ : \ y_1 < y_2\}$, $P^2 = \{(y_1, y_2) \in \mathbb{R}^2 \ : \ y_1 > y_2\}$ and $P^3 = \{(y_1, y_2) \in \mathbb{R}^2 \ : \ y_1 = y_2\}$. Then the smooth functions are $f^1$, $f^2$ and $f^3$. Clearly $f^1(y_1, y_2) = y_1$ in $P^1$, so that $\nabla f^1(y_1, y_2) = (1, 0)$ in $P^1$. For similar reasons, $\nabla f^2(y_1, y_2) = (0, 1)$ in $P^2$.

The critical situation concerns $P^3$. The construction principle (8) yields that $y_1 = y_2 = f^3(y_1, y_2)$ in $P^3$. It can be written $y = f^3(y, y)$ for all $y \in \mathbb{R}$. The function $f^3$ being continuously differentiable, one has necessarily $1 = \partial_{y_1} f^3(y, y) + \partial_{y_2} f^3(y, y)$ for all $y$, that is

$$1 = \partial_{y_1} f^3(y_1, y_2) + \partial_{y_2} f^3(y_1, y_2) \text{ on } P^3. \tag{24}$$

Let us now examine what is the meaning of the modified chain rule formula which replaces the initial one (7). This modified chain rule formula can be taken from Theorem 2

$$1 = \widetilde{\nabla} S_1(f_0(x)) W_0 \text{ for all } x \in \mathbb{R}. \tag{25}$$

The key observations are that $W_0 = (1, 1)$ and that $\widetilde{\nabla} S_1(f_0(x)) = \nabla f^3(f_0(x))$ since $f_0(x) \in P^3$. Then (25) reduces to the identity (24) which holds by definition. So the paradox does not show up again.

**Remark 2.** *A simple geometrical interpretation emerges from the fact that (25) reduces to (24). Actually the gradient of $f^3$ can take any value in the direction* **normal** *to the line $P^3$ while it takes the correct value in the direction* **tangent** *to $P^3$.*

### 6.3 The Boustany example

This example is proposed in Boustany (2024) to exemplify the issues at stake with nonsmooth autodiff with *maxpoooling* functions. The example is directly implemented in PyTorch. One defines a first maximum function $\max_1$ for a vector $x \in \mathbb{R}^a$ of arbitrary size $a \geq 1$, together with a second maximum function $\max_2$ which is a PyTorch function. The scripts taken from Boustany (2024) are in Table 1. Then for given $x \in \mathbb{R}^a$, one defines the function $t \mapsto f(t) = \max_1(tx) - \max_2(tx)$.

```
def max₁(x):
res = x[0]
for i in range(1, a):
if x[i] > res:  res = x[i]
return res
```

```
def max₂(x):
return torch.max(x)
```

Table 1: Script of the functions $\max_1$ and $\max_2$

By construction $f$ is the null function.

However it is reported (Boustany, 2024, Table 1) that the derivative calculated with *autodiff* in PyTorch is not zero. More precisely take $x = (1, 2, 3, 4)$, then $f'(t)$ is (numerically) zero everywhere except at $t = 0$ where the derivative is $\approx -1.5$. Our own tests reported in Table 2 confirm this observation.

| $t$ | -1 | -0.5 | -0.01 | 0 | 0.01 | 0.5 | 1 |
|---|---|---|---|---|---|---|---|
| derivative of $f$ | 0 | 0 | 0 | -1.5 | 0 | 0 | 0 |

Table 2: Values of the derivative of $f$ calculated with autodiff within PyTorch

The explanation in the context of the Murat-Trombetti representation formula (9) is as follows. The function $\max_1$ calculated with *autodiff* is given in Table 1. It yields that the associated gradient of $\max_1$ is calculated accordingly to the decomposition of the space in Borel pieces

$$\begin{aligned}
P^1 &= \{x \in \mathbb{R}^4 | \ x_1 \geq \max(x_2, x_3, x_4)\}, & f^1(x) &= x_1, \\
P^2 &= \{x \in \mathbb{R}^4 | \ x_1 < x_2 \text{ and } x_2 \geq \max(x_3, x_4)\}, & f^2(x) &= x_2, \\
P^3 &= \{x \in \mathbb{R}^4 | \ \max(x_1, x_2) < x_3 \text{ and } x_3 \geq x_4\}, & f^3(x) &= x_3, \\
P^4 &= \{x \in \mathbb{R}^4 | \ \max(x_1, x_2, x_3) < x_4\}, & f^4(x) &= x_4
\end{aligned} \tag{26}$$

where $x = (x_1, x_2, x_3, x_4)$. Then the associated gradient is

$$\widetilde{\nabla}\max_1(x) = \ (1, 0, 0, 0) \text{ in } P^1, \quad (0, 1, 0, 0) \text{ in } P^2, \quad (0, 0, 1, 0) \text{ in } P^3, \quad (0, 0, 0, 1) \text{ in } P^4. \tag{27}$$

The representation that we propose for $\max_2$ is different. It is based on different Borel pieces

$$
\begin{aligned}
Q^i &= \{x \in \mathbb{R}^4 | \ x_i > \max(x_i)_{j \neq i}\}, & f^i(x) &= x_i, & i &= 1, 2, 3, 4, \\
Q^5 &= \{x \in \mathbb{R}^4 | \ x_1 = x_2 > \max(x_3, x_4)\}, & f^5(x) &= (x_1 + x_2)/2, \\
Q^6 &= \{x \in \mathbb{R}^4 | \ x_1 = x_3 > \max(x_2, x_4)\}, & f^6(x) &= (x_1 + x_3)/2, \\
Q^7 &= \{x \in \mathbb{R}^4 | \ x_1 = x_4 > \max(x_2, x_3)\}, & f^7(x) &= (x_1 + x_4)/2, \\
Q^8 &= \{x \in \mathbb{R}^4 | \ x_2 = x_3 > \max(x_1, x_4)\}, & f^8(x) &= (x_2 + x_3)/2, \\
Q^9 &= \{x \in \mathbb{R}^4 | \ x_2 = x_4 > \max(x_1, x_3)\}, & f^9(x) &= (x_2 + x_4)/2, \\
Q^{10} &= \{x \in \mathbb{R}^4 | \ x_3 = x_4 > \max(x_1, x_2)\}, & f^{10}(x) &= (x_3 + x_4)/2, \\
Q^{11} &= \{x \in \mathbb{R}^4 | \ x_1 = x_2 = x_3 > x_4\}, & f^{11}(x) &= (x_1 + x_2 + x_3)/3, \\
Q^{12} &= \{x \in \mathbb{R}^4 | \ x_1 = x_2 = x_4 > x_3\}, & f^{12}(x) &= (x_1 + x_2 + x_4)/3, \\
Q^{13} &= \{x \in \mathbb{R}^4 | \ x_1 = x_3 = x_4 > x_2\}, & f^{13}(x) &= (x_1 + x_3 + x_4)/3, \\
Q^{14} &= \{x \in \mathbb{R}^4 | \ x_2 = x_3 = x_4 > x_1\}, & f^{14}(x) &= (x_2 + x_3 + x_4)/3, \\
Q^{15} &= \{x \in \mathbb{R}^4 | \ x_1 = x_2 = x_3 = x_4\}, & f^{15}(x) &= (x_1 + x_2 + x_3 + x_4)/4.
\end{aligned}
\tag{28}
$$

It yields

$$
\widetilde{\nabla}\max_2(x_1, x_2, x_3, x_4) = \frac{1}{N(x_1, x_2, x_3, x_4)} (y_1, y_2, y_3, y_4)
\tag{29}
$$

where

- $N(x_1, x_2, x_3, x_4)$ is the number of $x_i$ $(1 \le i \le 4)$ equal to the maximum value,

- $y_i = 1$ if $x_i = \max(x_1, x_2, x_3, x_4)$, and $x_i = 0$ if $a_i < \max(x_1, x_2, x_3, x_4)$.

**Remark 3.** *The examination of (28) shows that $\widetilde{\nabla}\max_2$ is symmetrized. One has for example*

$$
\widetilde{\nabla}\max_2(\mathbf{0}) = \frac{1}{4}(1, 1, 1, 1), \qquad \mathbf{0} = (0, 0, 0, 0).
\tag{30}
$$

*This value corresponds to the piece $Q^{15}$ which is a line in a space of dimension 4.*

*More precisely, the associated gradient $\widetilde{\nabla}\max_2$ is invariant under the action of permutations among the values equal to the maximum. This is confirmed by numerical evidence based on elementary tests in PyTorch.*

Now let us calculate the associated derivative of $f$ at $t = 0$ with the rules of automatic differentiation $\widetilde{f}'(0) = \frac{d}{dt}\max_1(t, 2t, 3t, 4t)_{|t=0} - \frac{d}{dt}\max_2(t, 2t, 3t, 4t)_{|t=0}$ that is $\widetilde{f}'(0) = \widetilde{\nabla}\max_1(\mathbf{0}) \cdot (1, 2, 3, 4) - \widetilde{\nabla}\max_2(\mathbf{0}) \cdot (1, 2, 3, 4)$, where the notation of the associated derivative is justified using Lemmas 1 and 3. One obtains $\widetilde{f}'(0) = (1, 0, 0, 0) \cdot (1, 2, 3, 4) - \frac{1}{4}(1, 1, 1, 1) \cdot (1, 2, 3, 4) = 1 - 10/4 = -1.5$ which is the value reported in Boustany (2024) and in Table 2.

### 6.4 A simplified Boustany example

Finally we prepare another simple example in the spirit of Boustany (2024), but where the maximum is calculated with the *MaxPool1d* function of PyTorch. Maxpooling is an important and popular operation in modern neural networks.

We assemble the function $f(t) = \text{maxpool1d}(t, 4t) - \text{maxpool1d}(4t, t)$. Some values implemented in PyTorch are given in Table 3. The explanation of the value of $\widetilde{f}'(0)$ is as follows. Actually all numerical test show that the PyTorch function maxpool1d = torch.nn.MaxPool1d(2, $stride = 1$) with a window of 2 elements has the same associated gradient as the function $\max_1$ presented in Table 1. Therefore $\widetilde{\nabla}\text{maxpool1d}(0, 0) = (1, 0)$. Then $\widetilde{f}'(0) = (1, 0) \cdot (1, 4) - (4, 1)(1, 0) \cdot (1, 4) = -3$ which is the value in Table 3 observed in the numerical tests.

## 7 Conclusion

The Murat-Trombetti Theorem provides a simple functional framework which allows to manipulate composition of Lipschitz-continuous and piecewise-$C^1$ functions. It constructs an associated gradient which is defined

| $t$ | -1 | -0.5 | -0.01 | 0 | 0.01 | 0.5 | 1 |
|---|---|---|---|---|---|---|---|
| derivative of $f$ | 0 | 0 | 0 | -3 | 0 | 0 | 0 |

Table 3: Values of the derivative calculated with autodiff within PyTorch

for all values of the input variable. An associated gradient is not unique nevertheless. We have observed that the gradient obtained from nonsmooth autodiff in PyTorch is systematically equal to an associated gradient in the sense of Murat-Trombetti. This approach also provides a non ambiguous chain rule formula, that was actually the key in the original paper Murat & Trombetti (2003). Then we defined the framework of Lipschitz-continuous and piecewise-$C^1$ functions which is an associative algebra for the composition of functions. Connections with the method of breakpoints described in Daubechies et al. (2019) is a priori possible. We mention some problems which could be the subject for future research. Some of them are already evoked in Berner et al. (2019).

**Evaluation of the Lipschitz constant of a function modeled with Neural Network:** The numerical evaluation and use of the Lipschitz constant of a given Neural Network function where the weights $W_r$ and the biaises $b_r$ are given has the subject of recent research (Combettes & Pesquet, 2020; Virmaux & Scaman, 2018; Pintore & Després, 2024; Béthune, 2024). More solid foundations for these works can be obtained with associated gradients.

**More variables and training:** The associated gradient has been introduced and studied in this work on examples with limited number of layers and with limited number of variables. In practice a Neural Network function is defined with respect to space variables (typically $x$ in (1)) and to parameters (typically $W_r$ and $b_r$ in (1)). Then it raises the mathematical question of the definition of an associated gradient with respect to all variables. There is major practical interest in designing an associated gradient with respect to the parameters $W_r$ and $b_r$ only. This has been evoked in Section 5.2. It can be used to describe a functional setting for training sequences and to compare with the numerical tests in Boustany (2024).

**SciML:** SciML is a new discipline that seeks to use ML for solving PDE (partial differential equation) models that show up in physics and engineering (Klawonn & al., 2024, EMS TAG), (Després et al., 2024, is a recent event). An important example is PINNs (Physically Informed Neural Networks), we refer to Mishra & Molinaro (2023) and references therein. It is reasonable to foresee that some integral formulations used in SciML can be justified with Theorem 1.

### Acknowledgments

This work received fundings by *Agence Nationale de la Recherche*, program France 2030, reference ANR-23-PEIA-0004.

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

# A   A self contained proof of the Stampacchia property

The Stampacchia property Stampacchia (1963); Kinderlehrer & Stampacchia (2000) states that a function $w \in \mathrm{Lip}(\mathbb{R}^a)$ is such that

$$\nabla w(x) = 0 \quad a.e. \quad x \in \{x \in \mathbb{R}^a \ : \ w(x) = 0\}. \tag{31}$$

A simple proof comes from a regularization technique. See also Evans (2018).
**First regularization.** Consider $x \mapsto |x|_\varepsilon = \sqrt{x^2 + \varepsilon}$ for $\varepsilon > 0$. The derivative is $\frac{d}{dx}|x|_\varepsilon = \frac{x}{\sqrt{x^2+\varepsilon}}$.

Thanks to the Rademacher (1919) Theorem, $w$ admits a gradient $\nabla w \in L^\infty(\mathbb{R}^a : \mathbb{R}^a)$. Also $|w|$ admits a gradient $\nabla |w| \in L^\infty(\mathbb{R}^a : \mathbb{R}^a)$ as well because $|w|$ is also Lipschitz. Take a vectorial smooth test function with compact support $\varphi \in C_0^1(\mathbb{R}^a : \mathbb{R}^a)$. The integration by part formula holds

$$\int \nabla |w|(x) \cdot \varphi(x) dx = -\int |w|(x) \nabla \cdot \varphi(x) dx = -\lim_{\varepsilon \to 0^+} \int |w|_\varepsilon(x) \nabla \cdot \varphi(x) dx.$$

There is no difficulty in passing to the limit because $w$ is continuous. A reverse integration by parts shows that

$$-\int |w|_\varepsilon(x) \nabla \cdot \varphi(x) dx = \int \nabla |w|_\varepsilon(x) \cdot \varphi(x) dx = \int \frac{w(x)}{\sqrt{w(x)^2+\varepsilon}} \nabla w(x) \cdot \varphi(x) dx$$
$$= \int_{w(x)>0} \frac{w(x)}{\sqrt{w(x)^2+\varepsilon}} \nabla w(x) \cdot \varphi(x) dx + \int_{w(x)<0} \frac{w(x)}{\sqrt{w(x)^2+\varepsilon}} \nabla w(x) \cdot \varphi(x) dx + \int_{w(x)=0} \frac{w(x)}{\sqrt{w(x)^2+\varepsilon}} \nabla w(x) \cdot \varphi(x) dx.$$

In the right hand side, the last integral vanishes of course. In the first integral one has the boundedness $\left| \frac{w(x)}{\sqrt{w(x)^2+\varepsilon}} \nabla w(x) \cdot \varphi(x) \right| \le |\nabla w(x) \cdot \varphi(x)|$ where on the right hand side the function $x \mapsto |\nabla w(x) \cdot \varphi(x)|$ defines a function in $L^1(\mathbb{R}^a)$. One also has pointwise convergence almost everywhere with respect to $x$ $\frac{w(x)}{\sqrt{w(x)^2+\varepsilon}} \nabla w(x) \cdot \varphi(x) \to \mathrm{sign}(w(x)) \nabla w(x) \cdot \varphi(x)$ a.e. $x$. Therefore the Lebesgue dominated convergence Theorem yields

$$\lim_{\varepsilon \to 0^+} \int_{w(x)>0} \frac{w(x)}{\sqrt{w(x)^2+\varepsilon}} \nabla w(x) \cdot \varphi(x) dx = \int_{w(x)>0} \nabla w(x) \cdot \varphi(x) dx.$$

Similarly $\lim_{\varepsilon \to 0^+} \int_{w(x)<0} \frac{w(x)}{\sqrt{w(x)^2+\varepsilon}} \nabla w(x) \cdot \varphi(x) dx = -\int_{w(x)<0} \nabla w(x) \cdot \varphi(x) dx$. It yields the formula

$$\int \nabla |w|(x) \cdot \varphi(x) dx = \int_{w(x)>0} \nabla w(x) \cdot \varphi(x) dx - \int_{w(x)<0} \nabla w(x) \cdot \varphi(x) dx. \tag{32}$$

**Second regularization.** Let us now redo the calculation but starting from a different regularization of the absolute value. We take $x \mapsto |x|^\varepsilon = \sqrt{\left(x + \sqrt{\varepsilon}\right)^2 + \varepsilon}$ for $\varepsilon > 0$, with derivative $\frac{d}{dx}|x|^\varepsilon = \frac{x+\sqrt{\varepsilon}}{\sqrt{(x+\sqrt{\varepsilon})^2+\varepsilon}}$.

One has $\int \nabla |w|(x) \cdot \varphi(x) dx = -\int |w|(x) \nabla \cdot \varphi(x) dx = -\lim_{\varepsilon \to 0^+} \int |w|^\varepsilon(x) \nabla \cdot \varphi(x) dx$ and

$$-\int |w|^\varepsilon(x) \nabla \cdot \varphi(x) dx = \int \nabla |w|^\varepsilon(x) \cdot \varphi(x) dx = \int \frac{w(x)+\sqrt{\varepsilon}}{\sqrt{(w(x)+\sqrt{\varepsilon})^2+\varepsilon}} \nabla w(x) \cdot \varphi(x) dx$$
$$= \int_{w(x)>0} \frac{w(x)+\sqrt{\varepsilon}}{\sqrt{(w(x)+\sqrt{\varepsilon})^2+\varepsilon}} \nabla w(x) \cdot \varphi(x) dx + \int_{w(x)<0} \frac{w(x)+\sqrt{\varepsilon}}{\sqrt{(w(x)+\sqrt{\varepsilon})^2+\varepsilon}} \nabla w(x) \cdot \varphi(x) dx$$
$$+ \int_{w(x)=0} \frac{w(x)+\sqrt{\varepsilon}}{\sqrt{(w(x)+\sqrt{\varepsilon})^2+\varepsilon}} \nabla w(x) \cdot \varphi(x) dx$$
$$= \int_{w(x)>0} \frac{w(x)+\sqrt{\varepsilon}}{\sqrt{(w(x)+\sqrt{\varepsilon})^2+\varepsilon}} \nabla w(x) \cdot \varphi(x) dx + \int_{w(x)<0} \frac{w(x)+\sqrt{\varepsilon}}{\sqrt{(w(x)+\sqrt{\varepsilon})^2+\varepsilon}} \nabla w(x) \cdot \varphi(x) dx$$
$$+ \frac{1}{\sqrt{2}} \int_{w(x)=0} \nabla w(x) \cdot \varphi(x) dx$$

The Lebesgue dominated convergence Theorem yields the same limit as before for the two first integrals. However the third integral remains. One obtains

$$\int \nabla |w|(x) \cdot \varphi(x) dx = \int_{w(x)>0} \nabla w(x) \cdot \varphi(x) dx - \int_{w(x)<0} \nabla w(x) \cdot \varphi(x) dx + \frac{1}{\sqrt{2}} \int_{w(x)=0} \nabla w(x) \cdot \varphi(x) dx. \tag{33}$$

**Final part of the proof.** Comparison of (32) and (33) yields $\int_{w(x)=0} \nabla w(x) \cdot \varphi(x) dx = 0$ for all $\varphi \in C_0^1(\mathbb{R}^a : \mathbb{R}^a)$. This is equivalent to the Stampacchia property (31) because the test function $\varphi$ is arbitrary.

