# OpenReview forum: "A functional framework for nonsmooth autodiff with {\it maxpooling} functions"
_TMLR — Accepted by TMLR_

### Review · Reviewer_rA3G · 2025-01-22

**Summary Of Contributions:**

Update [March 15, 2025] --- after the discussion, the revised version addresses all of my original concerns. I have updated my score to say the claims are supported and it is relevant to the TMLR audience.

----

The paper studies non-smooth autodiff, focusing on differentiating ReLU and max operations arising in neural networks, making a connection to the associated gradient perspective from Murat and Trombetti (2003). S2 provides examples for the ReLU and max functions. S3 summarizes the results from Murat and Trombetti (2003). S4 expands on the algebra of Lipschitz-continuous and C^1 piecewise functions. And S6 revisits the examples.

**Audience:**

Yes

**Broader Impact Concerns:**

No concerns

**Claims And Evidence:**

Yes

**Requested Changes:**

The work would be strengthened by experimental results showing where the associated gradients add a significant advantage to a standard neural network setting

**Strengths And Weaknesses:**

While the paper makes an interesting connection to the associated gradients from Murat and Trombetti, the paper lacks experimental demonstrations to validate the contribution. My perspective is that the connection, while interesting, is not sufficient on it's own without further connecting to the practical relevance of experimentally improving non-smooth autodiff.

The non-differentiability of ReLU and max operations in neural networks is well-known. In many cases, these non-smooth regions only occur in a measure-zero set that is still subdifferentiable and are therefore not a significant issue during training. The Boustany (2024) paper does a good job at experimentally showing cases where this does not hold, and could maybe provide an experimental starting point for the paper under submission..

---

> ### Comment · Reviewer_rA3G · 2025-03-15
> **Updated score, recommending accept**
>
> Dear authors,
>
> Thank you for the discussion and updated version with the further discussion of Clarke's derivative and potential applications. I've gone through everything here and have no significant remaining concerns with the paper, so I have recommended it for acceptance and updated my original review to state it meets the claims and is relevant to the TMLR audience. I would still be really interested in seeing the practical applications here (as followup work) because I can believe there is some practical gain to be made here --- however, I cannot easily see it because the most compelling examples only hold on a measure-zero set of weights that are never encountered in practice.
>
> Best regards,
> Reviewer rA3G

---

### Review · Reviewer_bcK4 · 2025-02-22

**Summary Of Contributions:**

The paper proposes a rigorous approach to nonsmooth automatic differentiation by applying the Murat–Trombetti Theorem to define an “associated gradient” for Lipschitz, piecewise-C^1 functions such as ReLU and max-pooling. The authors propose a mathematical framework in which:

1.	For Lipschitz, piecewise-C^1 functions, the authors define a surrogate gradient called "associated gradient", which remains well-defined at every point of the domain.

2.	A chain rule formula holds for these associated gradients, thereby resolving apparent contradictions that arise when we compose nonsmooth functions or differentiate them at points of nondifferentiability.

3.	Concrete examples illustrate how “standard” autodiff can yield paradoxical results—yet those are interpreted via associated gradients in the Murat–Trombetti setting.

4.	The authors also demonstrate that nonsmooth autodiff systematically yields one valid “associated gradient” in the sense of the theorem, thus offering a novel explanation for how famous frameworks (PyTorch, Tensorflow, JAX) handle partial derivatives at nondifferentiable points.

From a theoretical standpoint, the Stampacchia property, the Rademacher Theorem, and classical results on composition algebras of piecewise-C^1 functions all come together to show that the Murat–Trombetti formalism is both rigorous and practically relevant. Overall, this work provides new clarity on how to manage nonsmooth autodiff.

**Audience:**

Yes

**Claims And Evidence:**

Yes

**Requested Changes:**

- Using citation commands \citep when it is necessary and standard punctuation would help the paper read more smoothly.

- A brief discussion on how the “associated gradient” differs from or complements classical subgradient or Clarke gradient methods would clarify its place in nonsmooth optimization.

**Typos**

- Fix mismatches such as “The functions $f_i$ for $i = 1,\dots,L$" vs. the use of $f_0$ in other sections.
- Equation (1) suggests the nonlinear activation functions are $S_r$ for $r=1, \dots, \ell$. But Example 2.1 introduces $S_0$.
- “So the gradient of f is written as a matrice…” --> matrix
- 1966 Stampacchia (1963). I don't understand the 1966.
- piecewise $C^1$ --> piecewise-$C^1$
- researches --> research
- A space between Murat-Trombetti and Theorem in the openreview abstract.

**Strengths And Weaknesses:**

**Strengths**

1. The scope of the paper is clear. This kind of study is necessary when an entire community (Deep Learning) relies heavily on autodiff.

2. The adaptation of the Murat–Trombetti Theorem gives a natural framework for nonsmooth autodiff.

3. The illustrative PyTorch experiments confirm that widely adopted frameworks implicitly align with the proposed framework. This suggests immediate practical value in understanding “mysterious” derivative behaviors.

4. Because many activation and pooling functions are piecewise linear, the concept of an associated gradient extends naturally to various deep learning components.

**Weaknesses**

1. The current notations make it difficult to distinguish between the input $x$ and trainable parameters $\theta$. Defining $f(\theta, x)$ explicitly—as is standard in machine learning with an objective $\min_{\theta} f(\theta, x)$ would help. Equation (2) focuses on $\nabla_x f$, and it is unclear why gradients with respect to the weights are not shown. In Example 2.1, $w_0$ is treated like a weight, but the authors derive $\frac{d}{dx}f$; clarifying or showing $\nabla_{w_0} f$ could tie the example more closely to training.

2. It lacks a comparison with alternative nonsmooth optimization methods (e.g., Clarke subgradients, conservative gradients, generalized subdifferential, etc..). This would clarify how the “associated gradient” concept aligns with or differs from standard approaches to nonsmooth autodiff.

3. While the paper briefly mentions differentiating with respect to network parameters, it does not fully demonstrate how this approach applies to large-scale backpropagation in practical scenarios. More explicit examples or a short discussion illustrating parameter gradients in typical training loops would reinforce its relevance for the machine learning community.

4.  While generally well-written, a few changes could improve the clarity of the manuscript. The proposed changes are outlined below in the requested changes section.

---

### Review · Reviewer_jYJx · 2025-02-28

**Summary Of Contributions:**

The purpose of the paper is to provide a correct chain-rule formula for Lipschitz functions, which are piecewise $ C1$ functions.
First, the authors provide failures of the standard chain rule formula on Lipchitz piecewise $C^1$ functions, including Relu and maxpool functions. The authors then recall the Murat-Trombetti Theorem that provides a "correct" chain rule formula for this class of functions. The authors then show that the class of Lipchitz piecewise $ C1$ functions is an algebra for the composition operation. Finally, they provide an example of how to apply the Murat-Trombetti to machine learning-related functions.

**Audience:**

Yes

**Claims And Evidence:**

Yes

**Requested Changes:**

Clarity:
- page 1. what is the goal of this introduction? The into felt very unstructured to me and should be rewritten.
- page 2. I would recall Rademacher 1919 Theorem in a proper Theorem environment
- "give a meaning to the chain rule formula" this is too vague of a formulation
- page 2 Can you clarify the following? "For feed-forward functions, our notations imply that Ar (x) = Wr is constant with respect to x, so there is no difficulty."
- page 2 Can you clearly point toward the difficulties? "Nonsmooth autodiff with respect to the weights and biases leads to similar difficulties"
- The first example is not very convincing, the authors themselves admit. I would remove it for clarity
- Section 2.3, what do you mean by exactly "chain rule formula (2) has no clear interpretation"?.
- The second example is the main motivation of the paper, **this example should be much better polished**. The example is very simple but the current display makes it hard to understand. I would group some of the equations in math mode.
- Part 3 is very clear until Theorem 1 is included. The hints of the proof of theorem 1 are currently not clear enough for me. This should be entirely re-written (or moved to appendix).
- Is Section 4 an original contribution? This is very unclear to me.
- In Section 5, would it be possible to only present in detail the simplified Boustany example? **In order to avoid the 15 Borel pieces in equation 25?**

**Strengths And Weaknesses:**

Strength:
- The paper tackles an important topic, often overlooked in deep learning: how to differentiate non-smooth functions. Overall the examples are convincing, and the community could greatly benefit from these theoretical tools


Weaknesses:
- **The clarity**. The contribution of the paper is to present the Murat-Trombetti Thoreom to the machine learning community, however, in its current form, the manuscript cannot be understood by machine learning practitioners. I think the paper is very important and I will champion it, **but it will require massive changes in order to have value for the machine learning community**

---

> ### Author Response · Authors · 2025-03-01
> **answer to jYjx**
>
> Hi, here the answers to your questions. A revised version is online with modifications.
>
> -
> Comment by the rev:
> "The contribution of the paper... . I think the paper is very important and I will champion it, but it will require massive changes in order to have value for the machine learning community"
>
> -> Thanks for the general comment on the importance of this presentation of the Murat-Trombetti Theorem and its consequences.
>
> Concerning the other parts of the comment, I respectively copy what I have already answered to another rev,
> which is that  the official  editorial policy of the journal  contains
> "TMLR emphasizes technical correctness over subjective significance, to ensure that we facilitate scientific discourse on topics that may not yet be accepted in mainstream venues but may be important in the future."
> I did my best to  prepare this contribution exactly in this spirit.
>
> Also the notion of a  "machine learning practitioners" is not defined in the TMLR editorial policy, so I do not understand how to interpret this part of the comment.
>
> -
> page 1. what is the goal of this introduction? The intro felt very unstructured to me and should be rewritten.
>
> -> answer was already in the introduction: We hope that this discussion will contribute to popularize
>      this approach among the Machine Learning scientific community and to show its potential for the description
>      of many problems that intervene in nonsmooth automatic diﬀerentiation.
>      Now it is written in a paragraph to be more central.
>
> -
> page 2. I would recall Rademacher 1919 Theorem in a proper Theorem environment
> -> Done
>
> -
> "give a meaning to the chain rule formula" this is too vague of a formulation"
> -> Paragraph is rewritten.
>
> -
> page 2 Can you clarify the following? "For feed-forward functions, our notations imply that Ar (x) = Wr is constant with respect to x, so there is no difficulty."
> -> See modification with  more explanation.
>
> -
> page 2 Can you clearly point toward the difficulties? "Nonsmooth autodiff with respect to the weights and biases leads to similar difficulties"
> -> Actually another reviewer also asked for more development on this topic. See new version,  section 5, in particular subsection 5.2.
>
> -
> The first example is not very convincing, the authors themselves admit. I would remove it for clarity
> ->  Even if  I perfectly agree the example is borderline from the view point of what is in a computer, I have a different opinion.
> The reason is that  there just is nowhere in mathematical literature where a rule is proposed to multiply a function which is not defined by a number, even if this number is zero.
>
> -
> Section 2.3, what do you mean by exactly "chain rule formula (2) has no clear interpretation"?.
> -> Paragraph is rephrased as :"the chain rule formula (2) does not make sense because the matrix-valued functions in the right
> hand side are not defined in a non ambiguous manner".
>
> ------
> The second example is the main motivation of the paper, this example should be much better polished. The example is very simple but the current display makes it hard to understand. I would
> group some of the equations in math mode.
> -> Some equations are grouped in math mode.
>
> -
> Part 3 is very clear until Theorem 1 is included. The hints of the proof of theorem 1 are currently not clear enough for me. This should be entirely re-written (or moved to appendix).
> -> Unfortunately Theorem 1 is the core of the paper, and even its proof is interesting from a mathematical standpoint and not so popular.
> You can notice that the proof is half of a page, so it is not long. I agree that some notions are subtle ones, but the reasons these notions are difficult to understand at first sight are somehow related to the reasons why
> nonsmooth autodiff coupled with  backprop is difficult to conceptualize.
> So I prefer this format.
>
> -
> Is Section 4 an original contribution? This is very unclear to me.
> -> Yes it is fully original as well as Theorem 2, and as no equivalent in the literature (just compare with Section 6.4.2  Back-Propagation in an MLP is the famous book "Deep Learning" by Goodfellow et al.).
> More material on the comparison with the literature is in Section 5 because it was asked for by another reviewer.
>
> -
> In Section 5, would it be possible to only present in detail the simplified Boustany example? In order to avoid the 15 Borel pieces in equation 25?
> -> No, it is not possible, because the whole goal of the paper is  stated first line in the abstract and in the introduction:
> "We make a comment on the recent contribution Boustany (2024), ...."
> So it is needed in this paper to fully construct the solution to Boustany example.
> Without this example, it would be another paper.
>
> Additionally it is striking that the value that explains the result by Boustany corresponds to the piece $Q^{16}$ which is a line (dimension is 1) in a space of dimension 4.

---

### Author Response · Authors · 2025-01-28
**Answer to "Review of Paper3980 by Reviewer rA3G"**

It seems to me that the evaluation misses two important features, both on  mathematical theory and numerical evidence.
Let me make my point.

----------------------------------
Mathematical theory:
The theory of sub-differentials (as you mentioned it) is restricted to convex functions because it comes from optimization theory. So it cannot be adapted to composition of Lipschitz functions because deep Neural Network functions fail to be convex in the general case. Therefore I understand that what you mentioned is the theory of "generalized gradients" à la Clarke (famous textbook: F. H. Clarke, Optimization  and nonsmooth analysis, SIAM). The goal of Clarke's theory is to describe minimization of nonsmooth Lipschitz functions  which are possibly  non convex as well.  The derivatives of real valued functions are in general constructed as sets which are the convex hull of local natural  directions.

The point is that Clarke's theory is not able to describe the generalized gradient of composition of many Lipschitz functions with equalities, just embeddings or inclusions.
In Clarke's book, the 3 theorems on chain rule formulas are:
Section 2.3.9 Theorem (Chain Rule I) page 42,
Section 2.3.10 Theorem (Chain Rule II) page 45, and
Section 2.6.6 Theorem page 72 on Jacobian chain rules.
In case the resulting function is the composition of two general Lipschitz functions, none of these theorems is able to provide an equality, only embeddings of differential sets.

On the contrary, the Murat-Trombetti Th. is not focused on optimization, so  it does not rely  on differential inclusions of any kind. It is based on equalities (see Th. 1 in the paper under review, and Lemma 1 for example).
The explanation is that the Murat-Trombetti framework uses a strict sub-set of Lipschitz functions (piecewise C^1 functions), and it is very well adapted to the study of nonsmooth autodiff of piecewise C^1 functions.

------------------------------------
Numerical evidence:
Sub-differentials and generalized gradients express gradients as sets, and if the function is smooth, the set is just the classical derivative.
The Boustany examples are described in 5.3 (and 5.4) in the paper under review. By construction these two Boustany functions vanish identically.  Therefore one will inevitably obtains that  the differential set of a Boustany function 5.3 (or 5.4) at t=0 is {0}.

On the contrary, numerical evidence in PyTorch shows that autodiff calculates {-1.5} at t=0 for 5.3 and {-4} for 5.4.
The point is that these numerical values  are the direct consequence of the Murat-Trombetti algebra, which shows that the Murat-Trombetti framework perfectly explains what is observed in these elementary numerical experiments.

----------------------------------

Here I make additional remarks/answers.

The review asked  Requested Changes:
"The work would be strengthened by experimental results showing where the associated gradients add a significant advantage to a standard neural network setting."
My answer is that, for the reasons explained above, the standard neural network setting based of generalized gradients and sub-differentials is not able to explain the numerical values explained in basic Boustany tests
So the paper under review is already a a significant advantage to a standard neural network setting.

Claims and evidence: I do not understand the comment in the review which is "no", since all proofs are provided and numerical evidence is provided that the new framework is adapted to explain what is obtained in basic non smooth autodiff tests.

---

> ### Comment · Reviewer_rA3G · 2025-01-28
>
> Hi, thank you for the response! I'll be thinking about all of that. Here's a quick response for now:
>
> > The review asked Requested Changes: "The work would be strengthened by experimental results showing where the associated gradients add a significant advantage to a standard neural network setting." My answer is that, for the reasons explained above, the standard neural network setting based of generalized gradients and sub-differentials is not able to explain the numerical values explained in basic Boustany tests So the paper under review is already a a significant advantage to a standard neural network setting.
>
> I maintain the example is indeed interesting, but still disconnected from practice as is seems difficult to find a standard experimental setting training a network on a task where these issues impact the performance. Without these, the paper seems disconnected from practice.
>
> > Claims and evidence: I do not understand the comment in the review which is "no", since all proofs are provided and numerical evidence is provided that the new framework is adapted to explain what is obtained in basic non smooth autodiff tests.
>
> Continuing on the last point, I have responded no here because the paper studies the issue of non-smooth autodiff, which is a very practical and experimental setting, but the paper does not present significant evidence in any existing experimental non-smooth autodiff setting. The title states "non-smooth autodiff with maxpooling functions" -- max pooling is a widely deployed operation across many modeling tasks. I would be convinced if it's possible to show where an issue arises during training of a model due to the non-smoothness, and how to fix that. At the same time, I believe it will be difficult to show as some of the non-smoothness issues only arise on a measure-zero set of points that are not hit in practice, and if they are most frameworks return a valid (Clarke) subderivative.

---

### Author Response · Authors · 2025-02-01
**Answer to comments 28 Jan 2025**

Hi, thanks for the comments.
My response is organized as follows.

-------
-------
Point 1, answer to: "I maintain the example is indeed interesting, but still disconnected from practice as is seems difficult to find a standard experimental setting training a network on a task where these issues impact the performance. Without these, the paper seems disconnected from practice.".

Main point: The objectives of the paper are just to popularize a mathematical tool which seems powerful to explain/interpret the regularity and differentiability of functions encoded with non smooth activation and max-pooling functions. The paper never claimed to have direct "applicative" goals.


Minor point: What you refer to as a "standard experimental setting training" is related to  differentiability with respect to the weights of a Neural Network function, while  the article is more concerned with differentiability with respect to the input variable.
Differentiability with respect to input variable is of paramount importance for stability studies and  SciML,  which is  new field of application seeking to use ML for solving partial differential equations. In other words, this paper is not primarily concerned with training but with the  evaluation the regularity of the function obtained at the end of (any) training session.

Minor point: I respectively point out that the comment is not perfectly in line with the official editorial policy of TMLR https://jmlr.org/tmlr/, in particular the sentence " TMLR emphasizes technical correctness over subjective significance, to ensure that we facilitate scientific discourse on topics that may not yet be accepted in mainstream venues but may be important in the future".   Evaluation of  what we observe "in practice" is, for me, more a matter of personal and subjective significance.

-----------
-----------
Point 2, answer to: "I believe it will be difficult to show as some of the non-smoothness issues only arise on a measure-zero set of points that are not hit in practice, and if they are most frameworks return a valid (Clarke) subderivative.".

Minor point on wording: the Clarke derivative is not exactly a subderivative, for exemple the Clarke derivative of -|x| (minus the absolute value of x) at x=0 is the interval {-1,1}. So in this case, the Clarke derivative is more an upper derivative.

Main point: Example 5.2 in the article has exactly the structure of an academic oversimplified Convolutional Neural Network (one linear function composed with one maxpooling=max function).The beauty of this exemple relies on the fact that, even if the function "max" is differentiable everywhere except a set of measure zero, the composition with a linear function has the consequence that only the set of measure zero matters !!!!! So it is an academic example of the non-smoothness issues in NNs. The result of the composition is correct nevertheless thanks to Corollary 2 formula (14).

Minor point: One can compare (still for example 5.2) the Clarke's derivative and the associated gradient. The Clarke's derivative of the function "max" is given in Exemple 2.2.8 in Clarke's textbook: it is the set of all vectors which satisfy an equality constraint $\sum_i \xi_i=1$ with inequality constraints $\xi_i\geq 0$ and additional degeneracy constraints. The associated gradient of the same function "max" is one vector which satisfies an  equality constraint $\sum_i \xi_i=1$, with the same additional degeneracy constraints, but without any inequality constraint. So it is more general.

------------
------------
Point 3, additional consideration:

Once again the theory of Clarke's derivatives is not enough to exactly determine  composition of functions. For example take Corollary 2 page 39 in Clarke's book: to have an equality, it is needed that all but at most one functions are differentiable in the classical or strong  sense.
You can compare with Lemma 3 in the paper which expresses the additivity of associated gradients for general Lipschitz and piecewise C^1 functions without any such requirement, which explains why this approach is more descriptive and general.

---

> ### Comment · Reviewer_rA3G · 2025-02-01
>
> Thanks for the response again, they are reasonable points. I will take them all into consideration. The paper would be improved by integrating parts of this discussion into it on Clarke's subdifferential.
>
> I still feel that there is a disconnection between some parts of the response. For example, the authors say:
>
> > The paper never claimed to have direct "applicative" goals.
>
> And then next say:
>
> > Differentiability with respect to input variable is of paramount importance for stability studies and SciML, which is new field of application seeking to use ML for solving partial differential equations. In other words, this paper is not primarily concerned with training but with the evaluation the regularity of the function obtained at the end of (any) training session.
>
> This indicates direct applicative goals (e.g., numerically solving PDEs) and the most significant contribution would be to experimentally show it in a standard setting here.
>
> > Minor point: I respectively point out that the comment is not perfectly in line with the official editorial policy of TMLR https://jmlr.org/tmlr/, in particular the sentence " TMLR emphasizes technical correctness over subjective significance, to ensure that we facilitate scientific discourse on topics that may not yet be accepted in mainstream venues but may be important in the future". Evaluation of what we observe "in practice" is, for me, more a matter of personal and subjective significance.
>
> The editorial policy says technical correctness is emphasized over significance, but not that we should completely ignore discussing the significance or bringing up improvements that would strengthen the contribution. I still maintain that experimentally demonstrating the use of the associated gradient will significantly improve the contribution.
>
> >> I believe it will be difficult to show as some of the non-smoothness issues only arise on a measure-zero set of points that are not hit in practice, and if they are most frameworks return a valid (Clarke) subderivative.
>
> > Main point: Example 5.2 in the article has exactly the structure of an academic oversimplified Convolutional Neural Network (one linear function composed with one maxpooling=max function).The beauty of this exemple relies on the fact that, even if the function "max" is differentiable everywhere except a set of measure zero, the composition with a linear function has the consequence that only the set of measure zero matters !!!!! So it is an academic example of the non-smoothness issues in NNs. The result of the composition is correct nevertheless thanks to Corollary 2 formula (14).
>
> What the authors refer to as an "academic oversimplified Convolutional Neural Network" here in example 2/5.2 is simply the function $f(x)=\max(x,x)$, I think it is incorrect to refer to this as an oversimplified CNN as there are no non-trivial convolutions. The non-smooth part of the example only holds on a measure-zero set of the weights (I believe when $W_0=\alpha (1, 1)$ for $\alpha\in \mathbb{R}$). This function always hits the non-differentiable part of max because it just passes $x$ into it. Despite this, PyTorch's autodiff properly handles it:
>
> ```
> >>> import torch
> >>> x = torch.randn(42).requires_grad_()
> >>> y = torch.max(x,x)
> >>> y.sum().backward()
> >>> x.grad
> tensor([1., 1., 1., 1., 1., 1., 1., 1., 1., 1., 1., 1., 1., 1., 1., 1., 1., 1.,
>         1., 1., 1., 1., 1., 1., 1., 1., 1., 1., 1., 1., 1., 1., 1., 1., 1., 1.,
>         1., 1., 1., 1., 1., 1.])
> ```
>
> This perhaps corroborates the paper's claim "A general conclusion will be that the gradient constructed
> through nonsmooth autodiff in PyTorch is systematically equal to an associated gradient in the sense of Murat-Trombetti". Maybe the connection to the associated gradients here is interesting enough for an acceptance, but in the current form it seems like PyTorch and other autodiff frameworks were able to implement the non-smooth behavior in a reasonable way without the perspective of associated gradients, even though they seem to coincide. To me, the most valuable contribution would be to show a case where the associated gradient significantly improves the default behavior.

---

> > ### Comment · Reviewer_rA3G · 2025-02-03
> >
> > I was thinking more about this example and realize I wrote the incorrect measure-zero set of weights above. I just edited my comment there and corrected it, and it does not change any of my main points.
> >
> > Also using this example, I want to try to articulate again why I think some claims of the paper are not held. The authors write:
> >
> > > [the example] has exactly the structure of an academic oversimplified Convolutional Neural Network. [...] the composition with a linear function has the consequence that only the set of measure zero matters !!!!! So it is an academic example of the non-smoothness issues in NNs
> >
> > I perceive it as overclaiming and borderline misleading to say it's an example of non-smoothness issues in NNs (in general) because the issue only holds on a carefully-selected measure-zero set of weights: this example reduces to the non-smoothness of differentiating $f(x)=\max(x,x)$. And I still maintain my previous comment that it is incorrect to refer to this function (or slight generalization of it with a carefully-selected set of weights) as a CNN. In practice for standard settings, this case will almost-surely never arise when learning the weights because the issue only arises on a measure-zero set of weights. And even if it does, then the existing standard non-smooth autodiff framework behavior seems to handle the case okay. Maybe there is some variation beyond this where non-smoothness issues arise more generally and are not as easy to overcome, but I do not easily see it. That would be extremely interesting to show.

---

### Comment · Reviewer_rA3G · 2025-02-01
**Related paper: Understanding Notions of Stationarity in Non-Smooth Optimization**

[This paper](https://arxiv.org/abs/2006.14901) studies non-smooth optimization via Clarke subdifferentials and also mentions the application to neural networks, and includes some theorems on the chain rule, and a pseudo chain rule in some examples where it doesn't hold. They also include a few examples using max functions. Could the authors please comment on how this paper relates to the paper under submission? It seems useful to also include along with a broader discussion of Clarke subdifferentials.

---

### Author Response · Authors · 2025-02-03
**Preparation of an improved version**

Hi,
before answering at best the last questions of the reviewer, I have question about the TMLR reviewing process.

-------------------------
Question about the TMLR reviewing process:

I am of course opened to continue the discussion so as to provide  all explanations needed by the rev.
However I do not see clear indication if ever 2nd version will be to be prepared  with detailed answers to first round of comments/remarks/questions,
I even  do not know if a 2nd version is to be prepared in a normal TMLR reviewing process.
Can you provide indications on such issues ?

--------------------------
Answer to technical questions:

 I think I have enough material to prepare a 2nd version of the paper which answers his last questions, in particular
- sound discussion of Clarke's differential in the context of the paper,
- references  to the paper "Understanding Notions of Stationarity in Non-Smooth Optimization" (proposed by the rev), in particular the section "Sum rule" which pinpoints the same exact difficulties with Clarke's derivatives (The failure of the sum rule is one of the obstacles to computing the Clarke subgradient).
- and finally to present some prescriptions obtained from the present paper (asked by  Reviewer rA3G 01 Feb 2025 at 17:28). It will not be the same network as the one proposed by the rev because there is no difficulty, but I hope it will be valuable since it can be compared with the notion of stationarity developed in the suggested reference.
- but as above, I do not know if I have to answer directly in this chat or if I have to prepare 2nd version for next round of evaluation.

I agree that the presentation of the case $x|->\max(x,x)$ is misleading, in the sense it is only an oversimplified situation which does not pretend to be representative of training "real" CNNs, where the set of exceptional configurations is necessarily of measure zero. The example cannot not represent the field in view of training of CNNs. Please note that I never said that.
Nevertheless I  must  make this crystal clear clear in the introduction.

---

### Author Response · Authors · 2025-02-07
**answer to comment 01 Feb 2025 at 17:26**

Hi,

the provided piece of code is not representative of the phenomenon studied in the paper.
I propose the following one which  is more representative.
You can check that the final gradient evaluated with respect to s=0 depends on the ordering of the tensor.
Of course, this effect is  visible only a set of measure zero (no issue with s=2 for example),
in accordance with the common sense and with Th. 1.

-------------------------------
-------------------------------
 s=0.

t=torch.tensor(s).requires_grad_()

input_tensor = torch.tensor( [ [1, -6], [3, 4] ], dtype = torch.float32)

input_tensor = input_tensor.reshape(1, 1, 2, 2)

y=t*input_tensor

pool = nn.MaxPool2d(kernel_size=2, stride=2)

output = pool(y)

output.backward()

grad1=t.grad


t2=torch.tensor(s).requires_grad_()

input_tensor_2 = torch.tensor( [ [-6, 1], [3, 4] ], dtype = torch.float32)

input_tensor_2 = input_tensor_2.reshape(1, 1, 2, 2)

y_2=t2*input_tensor_2

pool = nn.MaxPool2d(kernel_size=2, stride=2)

output_2 = pool(y_2)

output_2.backward()

grad2=t2.grad



print (grad1,grad2)

---

### Decision · Action_Editor_CinU · 2025-04-24

**Recommendation:** Accept as is

**Comment:**

This paper provides a theoretical framework for nonsmooth automatic differentiation using the Murat–Trombetti theorem, introducing the notion of an “associated gradient” that remains valid at nondifferentiable points. The reviewers recommended acceptance, highlighting the work’s relevance on understanding how deep learning frameworks handle nonsmooth functions like ReLU and max pooling. Suggestions were made to improve clarity and accessibility for a broader ML audience, the authors have addressed key concerns.
The paper meets TMLR std for correctness and scientific value.

**Audience:**

Although some reviewers note that the paper's presentation could be improved to reach a broader set of machine learning practitioners, the content is clearly of relevance to the TMLR community.

**Claims And Evidence:**

The claims made in the submission are supported by mathematically rigorous evidence. The authors correctly identify an issue in automatic differentiation of nonsmooth functions, and use the Murat–Trombetti theorem to define and formalize an “associated gradient” that remains valid even at nondifferentiable points. This is demonstrated through mathematical arguments and concrete examples.